# Peer-led interventions: Exploring the peer group leader experience of delivering Sauti ya Vijana, a group-based mental health intervention for youth living with HIV in Tanzania

Chinenye Agina[1]*, Fortunata Nasuwa[2], Justina Mosha[3], Nasra Abdul[3], Erica Sanga[3], Leila Samson[4], Liness Amos Ndelwa[5], Blandina T. Mmbaga[1,2,6,7], Joy Noel Baumgartner[8☙], Dorothy E. Dow[1,2,9☙]

**1** Duke Global Health Institute, Duke University, Durham, North Carolina, United States of America, **2** Kilimanjaro Christian Medical Centre, Moshi, Tanzania, **3** Mwanza Research Centre, National Institute of Medical Research, Mwanza, Tanzania, **4** Ifakara Health Institute, Ifakara, Tanzania, **5** Mbeya Zonal Referral Hospital, Mbeya, Tanzania, **6** Kilimanjaro Clinical Research Institute, Moshi, Tanzania, **7** Kilimanjaro Christian Medical College University, Moshi, Tanzania, **8** School of Social Work, University of North Carolina at Chapel Hill, Chapel Hill, North Carolina, United States of America, **9** Department of Pediatrics, Infectious Diseases, Duke University Health System, Durham, North Carolina, United States of America

☙ DD and JNB are Joint Senior Authors
* chinenye.agina@duke.edu

## Abstract

Youth living with HIV (YLWH) face mental health challenges which negatively influence their adherence to antiretroviral medication and HIV outcomes. In Sub-Saharan Africa, where the majority of YLWH reside, there are few mental health professionals. Task-shifting interventions to lay peer leaders may be an effective strategy for addressing mental health challenges. This study aims to understand and evaluate peer group leaders' experiences delivering a peer-led, group-based mental health intervention called The Voice of Youth (Sauti ya Vijana [SYV] in Swahili) to YLWH in Tanzania. Peer group leaders (PGLs) aged 23–29 years at the time of hire and living with HIV were trained to deliver SYV. The study took place across four regions in Tanzania. In depth interviews (IDIs) (N = 25) were conducted in 2023 with PGLs after delivering the scaled pilot test of SYV. IDIs were audio-recorded in Swahili and translated and transcribed into English. Thematic analysis was applied using NVivo for coding and Excel to further summarize data and identify themes. PGLs experiences are organized via two CFIR domains:.Individuals Involved and Inner Setting. Under the Individuals Involved domain, PGLs described motivations such as a desire to help youth, increased confidence, and shared personal growth, as well as emotional challenges related to youth trauma, and how they thought community members perceived them. Within the Inner Setting domain, PGLs highlighted collaboration and collegial support alongside challenges related to social dynamics, compensation, and supervision. Across themes, PGLs emphasized sustainability, offering recommendations

**Data availability statement:** Data and related metadata underlying the findings are all qualitative in nature. The data codebook will be deposited in the AMANI Dataverse repository (https://dataverse.unc.edu/dataverse/amani). Individual indepth interview transcripts will be provided based on the limitations of our data transfer agreement between Duke University and the National Institute of Medical Research in Tanzania.

**Funding:** This work was supported by the National Institutes of Health (R01MH124476 to DD; Duke Global Health Summer Project Pilot funds to CA). The funders played no role in study design, data collection, data analysis, manuscript preparation, nor the decision to publish.

**Competing interests:** The authors have declared that no competing interests exist.

to strengthen program expansion and long-term impact. Insights from the PGLs can help enhance and position SYV for sustainability as Tanzania navigates scaling mental health care YLWH and also inform other peer-led mental health interventions in low-resource contexts.

## Introduction

### Global view of Youth living with HIV (YLWH)

Globally, the prevalence of human immunodeficiency virus (HIV) remains a pressing public health concern. In 2022, there were 3.2 million youth (15–24 years old) living with HIV (YLWH) and 350,000 new HIV infections occurred in this age group alone [1]. In 2019, two out of seven new HIV infections globally were among young people [2]. This issue is especially pressing in sub-Saharan Africa where 85% of adolescents (10–19 years old) who live with HIV reside [3]. Young adulthood is a significant developmental transition, and a large-scale meta-analysis of 192 epidemiological studies found that 62.5% of individuals with a mental disorder experience symptom onset before the age of 25 [4]. YLWH face increased mental health challenges compared to their HIV-negative peers, partly due to socioeconomic factors, HIV-related stigma, discrimination, and medication adherence challenges [5–7]. Addressing the mental health needs of YLWH is paramount [8].

Tanzania, with an annual population growth rate of 3.2%, predicts 140 million people by 2050; youth (15–24 years) made up 19.2% of the 65 million total population in 2022 [9,10]. In 2023, there were an estimated 150,000 youth (15–24 years of age) living with HIV and a staggering 19,000 new HIV infections [11]. A gender disparity is evident, with more than twice the prevalence among adolescent girls and young women living with HIV (1.3%) compared to adolescent boys and young men (0.6%) [12].

### Mental health among YLWH in Tanzania

The prevalence of depressive symptoms among Tanzanian adolescents living with HIV (age 7–18 years) was 27% in one local study conducted in a rural district of Tanzania, compared to 6% of age- matched adolescents who were HIV-negative using the Children Depression Inventory II [13]. Another study found 12% of YLWH in Tanzania reported symptoms of depression (score ≥ 10) based on the Patient Health Questionnaire-9 [14]. Such studies signify the critical need for increased mental health care among YLWH; and the need for interventions that address this population in Tanzania [6,14]. Tanzania's mental health workforce comprises of 1.31 professionals per 100,000 population. In child and adolescent mental health services, this number is 0.22 per 100,000 population [15]. During such a crucial point in their life, support from para-professionals could help youth cope with mental health challenges referring more critical cases to trained professionals [8,16]. YLWH need increased mental health resources that are likely to improve antiretroviral therapy (ART) adherence and virologic suppression, bolster their social support networks, and ultimately empower them to lead healthier lives [6].

## Peer-led Interventions

Peer-led interventions involve peer group leaders (PGLs) who help deliver interventions to a specific demographic. Peer-led interventions can help address the shortage of mental health professionals in low and middle income countries and help combat the mental health burden for YLWH [17–23]. PGLs close in age and with similar identities and lived experience to youth can provide relatable support and guidance that engenders trust and relatability on a deeper level than trained professionals [24]. PGLs are especially important in contexts like Tanzania where there is a lack of trained mental health professionals and prevalent unmet mental health needs [25,26]. There is evidence that peer support among YLWH can increase their self-esteem and empower them to make positive decisions about their health [27]. A study in Zimbabwe utilizing peer counselors found notable improvements in psychological well-being, quality of life, and adherence to ART for adolescents [16]. Adolescents in Tanzania who independently discovered their HIV status reported feelings of betrayal, which correlated with increased mental health symptoms of depression, post-traumatic stress, and adherence challenges compared to those who were purposefully informed [28]. A similar link between lack of disclosure and poor mental health and adherence has been shown in Zimbabwe [29]. Comfort levels regarding disclosure among YLWH also varied: some chose to conceal their status from peers, while others disclosed to friends and potential partners. This variability highlights the additional need for disclosure education and support. Having a safe space for individuals to discuss their HIV status with peers may help YLWH overcome disclosure challenges and support their mental health.

The primary objective of this study is to explore the experiences of PGLs delivering a mental health intervention to YLWH, Sauti ya Vijana (The Voice of Youth, SYV) as part of larger clinical trial examining effectiveness on HIV-related outcomes in Tanzania (NCT05374109). This exploration will help us understand the challenges PGLs face and identify factors they view as essential for sustaining a peer-led intervention.

## Methods

### Sauti ya Vijana (*The Voice of Youth, SYV*)

SYV is a peer-led, group-based mental health intervention for YLWH that aims to increase coping skills to manage common stressors and life challenges (Fig 1) [30]. It was co-designed in Tanzania with Tanzanian YLWH, a pediatric infectious diseases physician, and a clinical psychologist and incorporates components of evidence-based models to meet the needs YLWH described in our formative work [31]. SYV was co-created in 2016 and was evaluated in a pilot study in Kilimanjaro for youth 12–24 years of age [19]. SYV consists of 10 group sessions (two including caregivers) and two individual sessions. With the youth's consent, caregivers are encouraged to attend the first and sixth sessions. The first three sessions encourage youth participants to identify their stressors, worries, coping mechanisms, and learn components of cognitive behavioral therapy (CBT) to identify how their thoughts, feelings, and behaviors are connected [32]. Subsequently, youth learn about memories and the concept of the trauma narrative. In sessions four through six, youth have the opportunity to share their HIV disclosure narrative, what happened before, and after they learned their HIV diagnosis. This first occurs individually with a PGL. Then, if they are willing, youth are invited to share with the group in session five and with a caregiver or supportive adult who is already aware of their HIV status in session six. Components of Interpersonal Psychotherapy (IPT) [33,34] are used during session seven to encourage youth to identify their circles of support and to recognize who, if anyone, they to turn to for emotional support. Session eight focuses on recognizing stigma and how to cope with and reduce it using principles of CBT and HIV education. In session nine, youth discuss sexual reproductive health, engage in a condom demonstration, and learn about five steps to HIV disclosure, including role play. Session ten uses components of motivational interviewing to help youth identify their values and provide them with tools for behavioral change to move forward positively in their lives [35]. Youth then meet individually with a PGL to set future goals and reflect on what they have learned during SYV. There is a final content review and celebration where certificates are distributed to participants.

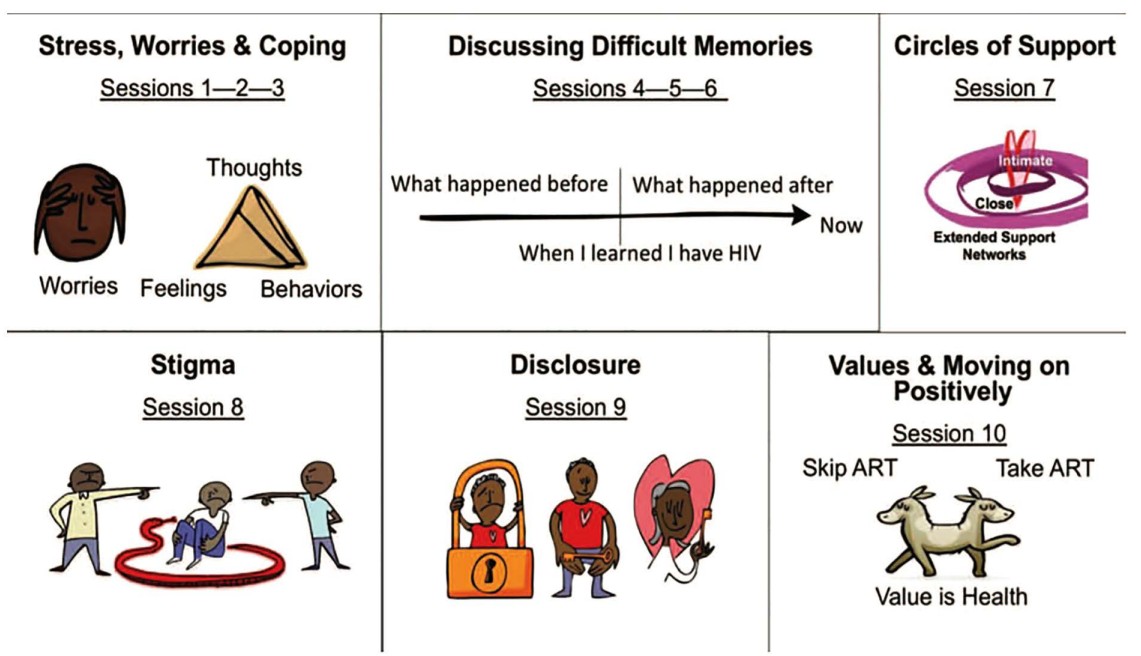

**Fig 1. Content of Sauti ya Vijana (SYV).** Sessions 1 and 6 involve caregivers; two individual sessions (after session 4 and 10) are not shown in Fig 1.

The head of the adolescent HIV clinic in each study region was approached to support recruitment of three male and three female PGLs for gender concordant group delivery. Suggested attributes included exhibiting excellent self-efficacy in managing their HIV care, leadership skills, ability to work well with youth and in a team, and confidence to create a safe space and effectively deliver the SYV intervention. Each site independently interviewed and hired PGL to deliver the SYV intervention. PGLs (N = 25) participated in an intensive two-week in-person training in Moshi, Tanzania to learn and practice the manualized group sessions prior to the scaled trial that included a pilot phase. After training concluded in November 2021, the PGLs continued practicing with their supervisors during the week and attended weekly supervision meetings. Due to Institutional Review Board (IRB) delays during the COVID-19 pandemic, PGLs had a full year of practice prior to the pilot phase of the trial. The SYV PGL role is a part-time role, taking place three days a week and includes a half day to practice intervention delivery, actual delivery of the intervention on Saturday morning, and writing of the session notes and fidelity checklist prior to session review and supervision the following week. The four sites offered various salary packages according to their unique institutional regulations, but all included competitive salary and health insurance benefits as part of compensation.

The SYV pilot phase of the trial was conducted from October to December 2022. During each session, two gender concordant PGLs facilitated the session and one PGL documented intervention fidelity using a fidelity check list and wrote session notes. These notes were reviewed by the PGLs and local supervision team each week. Supervision meetings were held in a hybrid format with PGL and the local supervisor meeting in person with expert group leaders and supervisors joining online via Zoom from offsite locations. For each regional site, supervision attendees included the PGLs, local supervisor, one or two expert group leaders (original SYV group leaders from the original 2016–2020 pilot that predated this trial and who helped train and supervise the new PGLs in this scaled trial), the expert supervisor (psychologist or psychiatrist), the study coordinator, and as able, the principal investigator [19,36]. The study coordinator facilitated the supervision meetings and took notes on the session progress, challenges, and recommendations for the upcoming sessions. SYV local supervisors were employed in SYV as a secondary role, all having a primary clinical roles within the HIV

or mental health clinic. Once a month, an all group leader Zoom meeting was held to review any challenges and receive didactic education on a mental health, life skills, or research topic.

## Setting

This study took place in Tanzania, a country located in East Africa with an overall HIV prevalence of 4.7% [1]. Study locations were chosen based on having a large population of YLWH, high mental health needs, an established adolescent HIV clinic, and space to conduct research. The sites include Moshi, Mwanza, Ifakara, and Mbeya, as shown with local HIV prevalence in Fig 2 [12].

## Participants

All PGLs involved in the SYV trial were invited to participate in this study (n = 25). PGLs were contacted by externally contracted qualitative researchers (not part of regular SYV study team) via phone to schedule a suitable interview time and location. PGL who agreed to participate met with the qualitative interviewer who obtained informed consent to conduct and record the in-depth interview. Participants received a sitting fee of 10,000 tsh (equivalent of $4.50). All PGL participated.

## Procedures

To ensure PGLs were free to express their true opinions without any influence on their job, their identifying information was kept confidential from the study's principal investigators. Four external qualitative researchers were hired, one at each site, to conduct the interviews, which took place from 10/02/2023–03/03/2023, following the completion of the pilot phase of the SYV trial in December 2022. To ensure privacy, the interviews were conducted at a private office or clinic space located on the grounds of the SYV-affiliated hospital for all four locations. A handheld audio recording device was used to

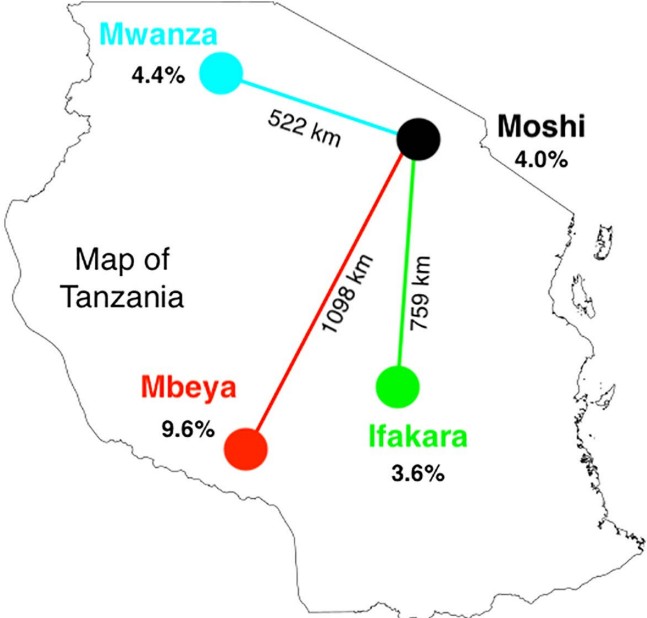

**Fig 2. HIV Prevalence in each SYV study region.** Base map layer sourced from Dreamstime (https://www.dreamstime.com/map-tanzania-outline-silhouette-vector-illustration-image135379273).

record, and the de-identified recorded interviews were uploaded to Duke Box, a HIPAA-compliant collaborative cloud storage platform used by Duke University. The qualitative researchers also took field notes during the interview and uploaded them using an electronic semi-structured online debriefing form on Duke Box. The in-depth interviews lasted anywhere from 60 minutes to 113 minutes, with an average duration of 74 minutes.

### In-depth interviews

The overall clinical trial is a hybrid implementation-clinical trial (ClinicalTrials.gov #NCT05374109) with a complementary qualitative study design informed by the Consolidated Framework for Implementation Research (CFIR) to ensure data collection from a variety of stakeholder perspectives [37]. In this manuscript we focus on the PGL perspective. The in-depth interview (IDI) guides for this current study are tailored in order to explore and understand the role of PGLs in the implementation of the SYV mental health intervention [38]. The IDI guide was created in accordance with the current study objectives of understanding the group leaders' motivation, experience, and identifying ways to improve SYV in the future. Most of the IDI guide questions thus fell under the CFIR domains of 'individuals involved' and 'inner setting'. Our thematic results are organized under these two domains with a cross-cutting theme of intervention sustainability. The topics under these domains included issues such as identity, motivations, experience as a group leader, group leader training, impact of intervention on PGLs, sustainability, and recommendations. The IDI guide included 32 open-ended questions (S2 Text).

### Demographic surveys

A short socio-demographic survey was given to PGLs to collect information on sex, age, marital status, whether they have children, other work (beyond SYV), location of other work, education level, and religion. This information was collected at the SYV in-person refresher training January 2024 and inputted into REDCap for storage and analysis.

### Data analysis

The IDI audio recordings were translated from Swahili to English. Each transcript was uploaded into HIPPA-compliant Duke Box and then imported into NVivo 12. A codebook was developed (see S1 Codebook), consisting of 28 deductive codes derived from the IDI guide plus one emergent code (economics). The first author (CA), based in the U.S., coded all 25 transcripts. To ensure the qualitative analysis was contextually grounded, a second Tanzanian qualitative researcher (FN) double-coded eight transcripts (~25% of all transcripts), two transcripts from each site, aiming to identity a similar interpretation of codes (34). No additional codes were identified during this process [38].

Thematic analysis followed a process of coding, reading coding reports, and creating an Excel matrix for data reduction and visualization purposes [39,40]. In addition, a summary document was created where each code had brief highlights of the responses. From this process, the first author coder was able to identify common themes and relevant quotes. The final compilation of themes was then presented to the study team for discussion at the in-person meeting January 2024 for PGL feedback. In our thematic analysis we found no sub-group differences based on gender. Previous studies indicate our sample size (n = 25) was more than sufficient for reaching data saturation [41,42].

### Ethical considerations

This study reports further details via the Consolidated Criteria for Reporting Qualitative Research (COREQ) (see S1 COREQ Checklist) [43]. Written informed consent was obtained from all PGLs. Ethical approvals were received by all IRBs including Duke University (Pro00109309), Kilimanjaro Christian Medical Centre (#2542), Ifakara Health Institute (IHI/IRB/EXT/No: 33–2023), Baylor Center of Excellence in Mbeya (SZEC-2439/R.E./V.1/27), and the Baylor Center of Excellence in Mwanza defers to the National Institute of Medical Research (NIMR/HQ/R.8c/Vol.I/2358). Additional information regarding the ethical, cultural, and scientific considerations specific to inclusivity in global research is included in the Supporting Information (S1 Checklist).

## Results

### Description of PGLs

All twenty-five PGLs consented to participation in the in-depth interviews. The ages of the PGL study participants at the time of interview ranged from 23 to 31 years old, with a median age of 26. The PGLs consisted of 12 women and 13 men, with seven at the Moshi site and six at each of the remaining sites: Ifakara, Mbeya, and Mwanza. Among them, 16 PGLs had children at the time of interview. All PGLs had completed at least primary education, with 14 completing secondary education and 8 having higher education. Outside of SYV, 13 PGLs had second jobs. Additional participant demographics can be found in (S1 Text).

### Themes under the "Individuals Involved" CFIR Domain

#### Desire *to* help youth *as* motivation *for* PGL role

A few PGLs had personal experiences with childhood peers who lost their lives to HIV, a driving force behind their decision to apply for the role. Due to their personal experiences, many expressed a general desire to help, with personal goals of youth empowerment. Additionally, PGLs relayed their interest in helping YLWH in managing depressive symptoms:

> *"I have seen many youth are going through depression and they don't have people to help them, and they are not aware of who they should take their problems to for help. Therefore, now as a group leader, I tell the youth which ways or methods they can use to get help now and in the future." (Female PGL)*

Throughout the intervention, the significance of providing a safe space for youth to express emotions became apparent to many PGLs. By providing this safe space, they were able to witness the youth's growth at the start and end of the intervention, evidenced by the youth participants knowledge and enthusiastic passion towards learning after each SYV session. Furthermore, PGLs valued their sincere interest to help youth and establish a heartfelt connection to support youth to give them hope for the future:

> *"I have a goal of teaching youth to understand about their general health. That living with [HIV] infection is not the end of life, you can have infection and have a family that is uninfected and live a good life like others." (Male PGL)*

#### Increased confidence *as a* perceived benefit

PGLs reported increased personal and work-related confidence attributing it to training, practice, and interactions with youth and co-workers. At the beginning of the intervention, a few PGLs stated that they did not feel confident prior to training. However, noticeable improvements in teaching skills, education, and passion for their role became apparent as confidence increased. As PGLs gained confidence, a few applied their acquired skills to their daily lives. For example, one PGL who often dealt with angry youth at the HIV care and treatment clinic (CTC) was able to use their leadership skills to de-escalate a situation. Several PGLs reported feeling respected in their communities, and a few even initiated educational groups in their local communities. PGLs noted the intervention's transformative impact on their personal lives, with a few PGLs shifting from having feelings of isolation and loneliness prior to undertaking the PGL role. For these individuals, the PGL role cultivated newfound confidence and self-expression. The shared experience of living with HIV provided reassurance, as PGLs found comfort in the knowledge that both their colleagues and participants of the intervention had also faced similar life challenges and events:

> *"At the beginning I felt despair because I was living with HIV infection, but when I got the opportunity to be with my fellow youth back there…it helped me … to believe that I am not alone…" (Male PGL)*

                                                                                                

### Recognition *of* shared personal benefit *with* youth

Some PGLs experienced a shared benefit while delivering the mental health intervention to the youth. Delivering the intervention became an empowering experience for PGLs as they taught and interacted with youth and observed transformations in the youth's behavior. One PGL emphasized the importance of reminding youth that living with HIV does not hinder one from being successful or setting long-term goals for their life. By instilling these values, youth were able to recognize one's own value is crucial for accomplishing personal objectives. Observing the impact of SYV on the youth helped PGLs understand how influential their work was. The process of developing new skills and earning a salary further motivated PGLs to plan for their future. For instance, two PGLs shared how working with SYV helped them shape their life plans:

> *"[Prior to being a PGL] I didn't have any goals at all...I didn't have any long-term goals, but after going through training and becoming a group leader, I changed my perspective and saw that I still have a long way to go. I was able to make long term goals and live a happy and healthy life" (Female PGL)*

During training, several PGLs found the connection between thoughts, emotions, and behaviors in CBT and the triangle [SYV sessions 1–3] beneficial in their personal lives. These sessions became valuable resources that many PGLs shared with close friends and family during conflict. Training and delivering sessions also had a positive impact on personal social support, particularly regarding disclosure, trust, and community [SYV Session 7–9]. These sessions helped PGLs accept their own situation by learning about stigma and identifying support networks in their lives. Simultaneously, a few PGLs acknowledged the necessity of prioritizing their mental health as they continued to discover the benefits of the intervention:

> *"Since I became a leader, I have become more compassionate and now I know a lot about mental health—it's something that helps me to cope with the environment at work, at home, and my life in general." (Male PGL)*

### Potential challenge *of* SYV sessions is *the* emotional impact *for* PGLs

Five PGLs shared their reflections on being reminded of past painful memories, an experience that led to secondary trauma. While training sessions proved fulfilling for many PGLs, handling the negative aspects of the sessions emerged as a significant challenge. One PGL expressed the emotional toll, stating:

> *"Hearing their stories can be painful in a way, but it also pushes me to find an angle that I can use personally to calm myself down. I wonder who I can talk to in order to ease the feelings that I will be carrying, and how I can show that I have been affected by the young person's problem, but it's me who is supposed to help them return to a normal mood." (Male PGL)*

During training, all PGLs were given the opportunity to discuss their own narrative of their journey with HIV to fellow leaders and supervisors. The PGLs demonstrated a high level self-awareness and were able to identify the unintended emotional challenges that arose during the sessions for them because of revisiting their past experiences.

### PGLs experienced mixed community perceptions *about* their role

The PGLs experienced diverse perceptions of their PGL role from individuals outside of the intervention. One PGL noted that some individuals assumed mental health issues in the PGL due to their involvement in a mental health intervention. Another PGL shared the experience of losing close friendships due to work commitments:

*"Well, before, I had friends with the same status as mine. Since I am now spending most of my time here, that means I am not seeing them often, so it is like they don't want to be with me anymore, but I also have new friends here--I mean my fellow group leaders. And when they [old friends] post their WhatsApp status it is like they are cutting me off, they are on their own, they don't want to post me, but it is okay" (Female PGL)*

One PGL mentioned that their job caused relatives to perceive them as overly proud due to the PGL having less time to visit them. Another found themselves feeling excluded due to their friends treating them differently while one PGL discovered instances of gossip circulating within their neighborhood:

*"Ah, some people in the neighborhood are speaking very negatively about me [his HIV status], especially those who come to this clinic. Whenever they come here, they see me around this clinic area…Some have even started spreading rumors in the community that I am not who they think I am. Because of this, I also feel very uncomfortable. However, based on the training I have received… I am trying to confront these negative feelings by using the techniques I've learned" (Male PGL)*

On the other hand, some PGLs found that their role brought them increased respect and recognition within their communities, with some even being referred to as 'nurse.' One PGL shared that people noticed positive changes in them, and recommended others seek their support in times of need. For these individuals, the PGL role fostered a sense of credibility and respect.

*"…I live with people on the street, and you can hear people say that I am now different from how I was before. Because in the beginning, there was a way I was living with people, and they thought my life was not proper—that my lifestyle was not good. But now, even if I pass somewhere, someone can say, 'Follow XXXX; he will be of help." (Male PGL)*

### Themes under the "Inner Setting" CFIR Domain

#### SYV experience helped foster collaborative relationships between PGL colleagues

The role of a PGL cultivated an environment that fostered collaborative relationships. Many of the PGLs worked well with their other PGL colleagues and viewed their relationships as a benefit of the role. The newfound relationships among PGL allowed them to forge new friendships, build a support system, and experience a family-like environment. A few PGLs even mentioned that before assuming their roles, they were very shy and had difficulty interacting with others:

*"I used to isolate myself…my life would be confined indoors. I mean, I didn't have those friends, I didn't know how to visit people or socialize in groups, I simply couldn't do it at all. So, this has helped me to interact with people. At least now I can even have a little conversation." (Female PGL)*

#### Social Dynamics *between* PGLs could also be challenging

Our research findings revealed that, despite developing connections with colleagues, several PGLs at three SYV sites encountered negative social dynamics. These encompassed gossiping, a sense of being underestimated, productivity issues, arrogance, and lack of cooperation from coworkers. For example, one PGL voiced frustration about issues being escalated instead of being resolved among themselves:

*"It's the gossiping that I hate—when someone takes things to higher-ups before we've even agreed to share it. It was supposed to stay between us. Small issues, and yet someone's already told the boss." (Female PGL)*

## Compensation concerns were common

PGLs frequently proposed raising the salary for their role, highlighting the financial challenges some faced in managing their current compensation. For example, one PGL suggested additional compensation for working outside of regular hours, such as visiting youth on weekends. One PGL reported hearing that other sites receive slightly higher pay, and that every site has its unique breakdown of payments. PGLs, acknowledging their dedication to ensuring youths comprehended the sessions, advocated for a higher salary. Furthermore, a PGL emphasized the elevated cost of living in Tanzania was high, noting that employees struggled to meet their expenses in the middle of the month:

*"At the middle of the month…when we are broke and when all the employees are crying that we don't have money, it will give you motivation because sometimes when we are broke, we lose moral …you find that you have nothing and you have long days to go…" (Male PGL)*

## Limitations *of* PGL Role

Maintaining boundaries with youth was a challenge for PGLs. While youth were very forthcoming with their problems, the PGLs were unable to solve them all due to role limitations:

*"Ok, the challenges of youth--when they have brought, maybe, complaints, and they need help, you may find that challenge, as a leader, is beyond my power. This is the challenge which as a leader makes you see that the work is so difficult." (Female PGL)*

The tasks and responsibilities of PGLs could be time consuming. One PGL stated that the role left them less time to spend with their children while at work on the weekends. A recognized limitation of the role for PGL was the recognition of the small number of PGLs at the site. When delegating tasks, responsibilities increased due to the limited number of staff to lead a group of youth.

*"The worries which I have are the small number of group leaders…. because there are tasks of collecting information, writing and reading, and now, who will be reading, observing the youths, and taking the information from the youths... There will be many things to do for one person to lead twelve people or ten people it is difficult." (Female PGL)*

## Challenges *of* teaching competence and cross-site PGL support and supervision

PGLs faced challenges regarding how the youth perceived their teaching abilities. Many voiced concerns that their inability to provide precise answers to questions might lead youth to question their knowledge. Moreover, a few PGL felt anxious about whether youth truly understood their teachings.

Another challenge for PGLs was the limited physical presence of supervisors. SYV supervision calls typically occur through video calls, and one PGL expressed reservations about openly discussing certain challenges in the presence of other PGLs. *"Yes, how can I say it on the call like we had a fight yesterday… Yes, we are all there, how do you think he will feel?" (Male PGL).* During supervision, PGLs are encouraged to address any issues that arose during sessions. One PGL recommended that colleagues from different sites should bring up issues earlier, rather than waiting to discuss them during these meetings.

PGLs discussed the challenge of not being able to hold physical meetings with other PGLs due to the intervention taking place at four different sites. For that reason, it was suggested to arrange in-person meetings to talk through their experiences, challenges, and foster learning with PGLs from all sites:

*"… We asked if we could meet with people from all the sites and be able to share the challenges that they have been going through and in what way they have been able to solve those challenges so that we get more understanding, because when you meet in one place and share with each other how to solve problems at least it sticks in the head quickly." (Male PGL)*

## Sustainability of SYV intervention

**Intervention Goals and Expansion.** When asked about the goal of the intervention, PGLs emphasized its purpose: to help youth open up about their lives. PGLs observed the impact the intervention had on the youth, and parents were appreciative of its influence on their relationships with their children. Many PGLs aimed to ensure the youth did not leave the intervention in the same state as when they came in:

*"As a leader of the group, what is important…is to ensure that those youth who are chosen to participate in SYV do not leave as they came and leave with something in their head, mostly they change completely, what we are going to teach them stays in their heads and they are able to implement and work on it. That is very important" (Male PGL)*

PGLs expressed an interest in broadening the reach and impact of the SYV intervention. To achieve this, a few PGLs recommended increasing the number of group leaders, utilizing radio or television for educational outreach, exploring schools for recruitment and session delivery, and extending teaching roles to different areas in Tanzania. One PGL articulated this aspiration by showing interest in helping youth nationwide with the intervention:

*"I want to teach about mental health in the whole of Tanzania in order to help youth. To spread SYV in all parts of Tanzania to reduce deaths, the deaths of children, who are stigmatized, those who do not take medications effectively, and the youths who get challenges" (Male PGL)*

**Career Advancement.** When asked about their desired duration as PGLs, many expressed wanting to continue in the PGL role for as long as possible, noting their role was not indefinite due to aging out or the project ending. Many were apprehensive about aging out of the peer group leader role:

*"My concern is about what I shall do after the end of this position as a leader, what am I going to do. Ehh, where am I going to get a job or in which situation am I going to be? That is the concern which I have right now" (Male PGL)*

Other PGLs described their aspirations to advance their career and secure higher-level positions in the future, stating "I will look for the opportunity to go for my studies" (Male PGL), while others showed a strong interest in careers in counseling. For instance, one PGL stated, "I would like to get more training on how to teach youths…" (Female PGL), and another added, "Develop myself educationally, study more counseling…" (Male PGL).

## Feedback and recommendations

PGLs all provided feedback to enhance the intervention for the future. The feedback included key group leader qualities: being a good leader, compassionate, cooperative, a problem solver, and having public speaking skills. PGLs also mentioned the importance of cross-site collaboration to share challenges and learn from others. Other recommendations included incorporating extra sessions for youth, increasing the frequency of in-person supervision meetings, and providing additional work computers. Moreover, a few PGLs suggested extending the duration of supervision meetings and introducing an in-person meeting with all supervisors to make these calls more beneficial:

*"… Well, these calls are good (hesitates) they are doing good though I can suggest that (short pauses) it would be better if those supervisors would be coming here physically more often to see things from the base... because if you aren't at the base (hesitates) I mean how sure are you that we have done what we have documented?" (Male PGL)*

A few PGLs strongly advocated for mixing girls and boys during youth sessions, emphasizing the potential for mutual learning to enhance the overall learning experience. One PGL proposed increasing the maximum age for youth participants would allow the intervention to reach a wider audience. The majority of PGLs also expressed that a salary increase would boost their motivation to work and provide crucial financial support to their families: *"…because prices of things in the markets have increased too. And if you look... prices have gone high and the business we depend on does not go well" (Female PGL).*

Some PGLs proposed additional training in counseling and psychology, along with periodic refresher sessions for group leaders to better equip them to deliver the intervention in the future. One suggestion involved incorporating an extra training session annually, while another PGL recommended distributing tests to assess PGLs comprehension of various topics. Another recommendation involved having expert group leaders observe PGLs practicing with the youth at each site:

*"I wish if the expert group leaders would come to see how we are practicing with the youth, because they only saw us practicing during training. You know during training we made so many mistakes because we just started, but after practicing we were okay, so I wish if the expert group leaders would visit us at our sites to see how we have improved" (Female PGL)*

The qualitative findings offer a diverse perspective on the task sharing experience for PGLs. Male and Female PGLs experienced similar benefits and challenges during their role. These results encapsulate perceived benefits, acknowledge challenges, and provide valuable recommendations for the intervention.

## Discussion

This research study aimed to explore the experiences of PGLs delivering a mental health intervention to YLWH. Our investigation revealed numerous motivators driving PGLs to participate in the intervention. The insights shared by PGLs highlight various aspects that contribute to the lay worker experience of delivering a manualized mental health intervention. Currently, there are limited studies on peer led mental health interventions [18–20,44–46], with only a few being delivered to YLWH in Africa [18,19,24,29,44,47].

The exploration of motivators for the PGL role in this study revealed a range of various experiences and values that drove their engagement in the role. A strong desire to positively impact the lives of youth consistently emerged as a key motivator. For example, one PGL shared how the experience of losing peers in their life motivated them to apply to the role, emphasizing how personal loss can shape career choices. Several PGLs also described a sense of responsibility not only to the youth in the intervention but also to peers in their broader community.

These findings align with previous studies showing that holding roles with meaningful impact on youth can be deeply fulfilling for PGLs [48], and that personal connections to the role often influence motivation and commitment [24,49]. In addition, prior research has found that shared experiences with the target population can enhance PGLs ability to relate to participants and foster engagement, which we observed in our study [44,50].

The shared benefit PGLs had while delivering SYV served as a motivator and benefit of their job. PGLs gave youth a safe space to discuss their challenges, which affected how the intervention was received and delivered. A 2022 systematic review found that one key facilitator for success in delivering a mental health intervention lies in lay providers' connection to the intervention, including positive attitudes, beliefs, behaviors, and intentions. Establishing relationships with their colleagues also played a crucial role, enabling PGLs to find a community with similar lifestyles. This support network helped

many of them cope with feelings of loneliness, particularly before starting the role. In addition to forming support networks, the skills gained by PGLs, for example, in Project YES! or the Wakakosha peer-led intervention allowed them to set personal and professional goals for the future [44,51].

While the delivery of the intervention to YLWH was personally rewarding, various barriers existed. The PGL role, serving as a newfound source of income for many, posed challenges in terms of compensation. This was due to rising economic costs, funding future career plans, and being responsible for providing for themselves and their families. Another study found that a few peer leaders contemplated quitting their roles to take up work with better pay due to their need to support their dependents [48]. An important factor to consider is standardizing wages across sites or adjusting salaries based on site-specific cost of living and promoting salary transparency across all sites.

The navigation of emotional discomfort emerged as a challenge for the PGLs. PGLs processed their own personal narratives during SYV training and found many youth shared similar traumatic experiences. Similarly, one study found that the PGL role often led to stress due to the painful experiences shared by youth, which in turn required the PGLs to identify coping strategies for themselves. Mentorship and peer support plays a crucial role in helping individuals deal with the stress of complex cases and stories [48]. In SYV, the supervisors and their fellow PGLs helped provide this support. Future scale up should emphasize mentorship and other support systems for PGLs to mitigate emotional discomfort caused by providing support to YLWH.

Support systems are essential to avoid burnout and stressors for PGLs. While delivering the SYV intervention, many PGLs encountered challenges related to youth issues outside of the boundaries of their training and role limitations. The unique connections formed with the youth posed difficulties when PGLs needed to maintain boundaries. Recognizing the need for supervision is essential in preparing PGLs to navigate such issues in the future. While the PGL role is successful at helping youth deal with their challenges, it introduces a dynamic that blurs the line between being a lay mental health provider and a friend. Several studies have explored similar themes where PGLs were not trained to deal with complex issues youth may face without reporting it to a supervisor. Though the issue of professional boundaries was discussed extensively in training, some of the challenges faced by PGLs underscore the need to continually focus on professional boundaries during supervision and ongoing training to ensure optimal team dynamics and support for youth. While PGLs are young adults who carry responsibility for youth, many entered the program through engagement with the clinics, often beginning as adolescent patients or peer educators. This background allows them to relate to participants in a meaningful way, but it also highlights the importance of careful screening and structured support to ensure they are fully prepared for the PGL role from the outset.

PGLs play a crucial role in supporting YLWH; however, they require critical support due to staff size, workload, and difficulties arising from delivering sessions and coping with painful memories [48]. As interventions conclude, PGLs face concerns about their career trajectory, given the short term nature of these interventions [52]. To address this challenge, studies suggest PGLs take on other roles in the clinic [53]. Another study found that lay workers expressed dissatisfaction with their high workload and relatively small salary, advocating for pay be comparable to that of other health professionals [48]. Given the unique lived experiences that PGL bring to their roles, it is essential that their compensation reflects this.

To ensure that PGLs are valued and that concerns about job security and inadequate compensation are addressed, it is essential to formalize their role within the healthcare system. The Tanzanian Ministry of Health, Community Development, Gender, Elderly, and Children (MoHCDGEC) can take the lead in this initiative by establishing a national program dedicated to the mental health of youth [15]. Additionally, the Tanzanian National AIDS Control Programme can expand its services to include peer-led mental health interventions for all Tanzanian youth living with HIV in routine HIV Care and Treatment Clinics [54]. These initiatives will not only add credibility to the PGL role, but also create opportunities for a broader range of roles in the current healthcare system, ultimately contributing to the retention of the mental health workforce [55]. To ensure a comprehensive approach, policy makers can synthesize research to form policies that translate mental health interventions into regionally run services in Tanzania [55]. Additionally, collaboration with local leaders and stakeholders will help forge long-term partnerships and foster the effective implementation of interventions.

The age requirement for PGLs, set between 23–30 years old, presents a challenge as leaders age out of their roles. The intervention's sustainability hinges on the transferability of the PGL skills to new PGLs. Aging out may result in the loss of experienced individuals who have developed deep relationships with youth participants. As older PGLs exit the intervention, new PGLs will have to be trained, prepared for their new job, and paid. This transition may result in the discontinuation of mentorship and support for the youth by the previous cohort of PGLs, as they move into roles that do not directly involve delivering the intervention. Preparing new PGLs will require extensive training and practice. Assigning former PGLs as trainers and supervisors for new PGLs, in a "train the trainer" model can mitigate knowledge transfer challenges. Use of digital technologies could support standardized training and ensure competence of new PGL cohorts.

All participants provided feedback and recommendations to enhance the SYV intervention in the future. PGLs emphasized the importance of certain qualities such as compassion, leadership skills, and problem solving in future PGLs to ensure the success of the intervention. Although participants had supervision sessions with their fellow PGL in person, the local supervisors would sometimes join virtually, along with the expert group leader and expert supervisors who were often in other cities. Two PGLs suggested that having local supervisors present in person for supervision would be more valuable. However, this approach would increase costs and affect sustainability, as supervisors currently have primary clinical roles and supervision is conducted in a hybrid format to accommodate staffing and site constraints. PGLs did not specify which supervisors should attend in person; however, occasional visits from expert supervisors could be helpful. Providing a dedicated space for discussion, supportive supervision becomes instrumental in addressing any challenges faced by PGLs [24,49,56,57].

This study contributes valuable insights to the current body of literature by offering insights into the firsthand experiences of PGLs engaged in the SYV task-sharing mental health intervention in Tanzania. Through in-depth interviews, our research provides a unique perspective on the motivators, challenges, and recommendations from the PGLs themselves. These perspectives can be integrated into the design of future mental health interventions for YLWH.

## Strengths and limitations

The strengths of the present study include the sample and study design. The PGLs, living with HIV, offer a unique perspective, understanding the uniqueness of what YLWH face. The utilization of in-depth interviews captured the authentic stories of PGLs, allowing PGLs to share their experiences freely due to the open-ended nature of the questions.

However, several limitations could have impacted the study results. PGLs were employees of the study and to reduce bias the qualitative interviewers were external to the project and study principal investigators remained blinded to information coming from an individual or site. Nonetheless, it is essential to consider the possibility of social desirability bias influencing PGLs responses. PGLs may have been influenced by concerns about job security or the confidentiality of their answers. The time lapse between the intervention training in October 2021 and the interviews conducted in February and March 2023 after delivering only one pilot phase intervention round is another crucial factor. Intervention delivery as part of the trial began April 2023, thus shared PGL experiences lead up to trial start. The close relationships among PGLs raises the possibility of them discussing interview responses with each other, which may influence the consistency of their answers. Although we believe our findings are transferable to similar settings in East Africa, they may not be transferable to different populations or settings.

This study contributes to the growing body of evidence that supports task-sharing mental health interventions across different contexts using PGLs [56]. The findings not only add to the vast knowledge derived from PGLs perspectives but also highlight areas for further exploration. Future research should focus on cost-effective methods to maintaining PGLs exploring train the trainer models and digital technologies as well as and implementing referral systems to support the PGL role [58]. Beyond the Tanzanian context, researchers should compare PGL experiences and potential applicability in different cultures [59]. PGL in this study were Tanzanian and cultural factors may influence intervention delivery in diverse contexts [50]. Current and future peer-led interventions will be better informed when incorporating the unique PGL's

perspectives on the specific needs and concerns of both their role and the YLWH they teach. Delving into these insights contributes to the existing literature on the role PGLs play in mental health care for youth.

## Conclusion

The experiences of the PGLs delivering the SYV mental health intervention to YLWH illustrates the impact of their work on both the youth and their own lives. In settings with limited resources, successful peer-led interventions can help bridge gaps in mental health professional availability. Considering the factors highlighted by PGLs can further enhance the SYV PGL experience and position SYV for sustainability as Tanzania navigates scaling mental health care for YLWH.

## Supporting information

**S1 Text. PGLs Description.**
(DOCX)

**S2 Text. Interview Guide.**
(DOCX)

**S1 Codebook. Codebook for In-depth interviews**
(DOCX)

**S1 Checklist: Inclusivity in Global Research Questionnaire.**
(DOCX)

## Acknowledgments

We extend our most sincere gratitude to all the participants for their time, insightful stories, and dedication to supporting YLWH through the SYV intervention. Additionally, we express thanks to the KCMC-Duke Collaboration and all of the participating SYV sites (Ifakara Health Institute, Baylor Center of Excellence Mbeya and Mwanza, KCMC and Mawenzi regional referral hospital), site supervisors, and other study support staff for their instrumental role in making this study and the SYV intervention possible.

## Author contributions

**Conceptualization:** Chinenye Agina, Dorothy E. Dow.

**Data curation:** Fortunata Nasuwa, Justina Mosha, Nasra Abdul, Erica Sanga, Leila Samson, Liness Amos Ndelwa.

**Formal analysis:** Chinenye Agina, Fortunata Nasuwa.

**Funding acquisition:** Chinenye Agina, Dorothy E. Dow.

**Investigation:** Fortunata Nasuwa, Justina Mosha, Nasra Abdul, Erica Sanga, Liness Amos Ndelwa.

**Methodology:** Chinenye Agina, Joy Noel Baumgartner, Dorothy E. Dow.

**Project administration:** Chinenye Agina.

**Resources:** Chinenye Agina, Blandina T. Mmbaga.

**Supervision:** Blandina T. Mmbaga, Joy Noel Baumgartner, Dorothy E. Dow.

**Visualization:** Chinenye Agina.

**Writing – original draft:** Chinenye Agina.

**Writing – review & editing:** Joy Noel Baumgartner, Dorothy E. Dow.

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
