## [Decision Letter · Decision Letter 0]

22 Apr 2025

PMEN-D-24-00520

Peer-led interventions: exploring the peer group leader experience of delivering a group-based mental health intervention for youth living with HIV in Tanzania

PLOS Mental Health

Dear Dr. Agina,

Thank you for submitting your manuscript to PLOS Mental Health. After careful consideration, we feel that it has merit but does not fully meet PLOS Mental Health’s publication criteria as it currently stands. Therefore, we invite you to submit a revised version of the manuscript that addresses the points raised during the review process.

We look forward to receiving your revised manuscript.

Kind regards,

Ibrahim Yigit

Academic Editor

PLOS Mental Health

Journal Requirements:

1. Please include a complete copy of PLOS’ questionnaire on inclusivity in global research in your revised manuscript. Our policy for research in this area aims to improve transparency in the reporting of research performed outside of researchers’ own country or community. The policy applies to researchers who have travelled to a different country to conduct research, research with Indigenous populations or their lands, and research on cultural artefacts. The questionnaire can also be requested at the journal’s discretion for any other submissions, even if these conditions are not met.  Please find more information on the policy and a link to download a blank copy of the questionnaire here: https://journals.plos.org/plosmentalhealth/s/best-practices-in-research-reporting. Please upload a completed version of your questionnaire as Supporting Information when you resubmit your manuscript. 2. We have noticed that you have a list of Supporting Information legends in your manuscript. However, there are no corresponding files uploaded to the submission. Please upload them as separate files with the item type 'Supporting Information'. 3. In the online submission form, you indicated that your data will be submitted to a repository upon acceptance. We strongly recommend all authors deposit their data before acceptance, as the process can be lengthy and hold up publication timelines. Please note that, though access restrictions are acceptable now, your entire minimal dataset will need to be made freely accessible if your manuscript is accepted for publication. This policy applies to all data except where public deposition would breach compliance with the protocol approved by your research ethics board. If you are unable to adhere to our open data policy, please kindly revise your statement to explain your reasoning and we will seek the editor's input on an exemption. 4. Some material included in your submission may be copyrighted. According to PLOS’s copyright policy, authors who use figures or other material (e.g., graphics, clipart, maps) from another author or copyright holder must demonstrate or obtain permission to publish this material under the Creative Commons Attribution 4.0 International (CC BY 4.0) License used by PLOS journals. Please closely review the details of PLOS’s copyright requirements here: PLOS Licenses and Copyright. If you need to request permissions from a copyright holder, you may use PLOS's Copyright Content Permission form. Please respond directly to this email or email the journal office and provide any known details concerning your material's license terms and permissions required for reuse, even if you have not yet obtained copyright permissions or are unsure of your material's copyright compatibility.  Potential Copyright Issues: Fig2.tif : Please confirm whether you drew the images / clip-art within the figure panels by hand. If you did not draw the images, please provide (a) a link to the source of the images or icons and their license / terms of use; or (b) written permission from the copyright holder to publish the images or icons under our CC-BY 4.0 license. Alternatively, you may replace the images with open source alternatives. See these open source resources you may use to replace images / clip-art:- https://commons.wikimedia.org https://openclipart.org/ 5. Figure 2: please (a) provide a direct link to the base layer of the map (i.e., the country or region border shape) and ensure this is also included in the figure legend; and (b) provide a link to the terms of use / license information for the base layer image or shapefile. We cannot publish proprietary or copyrighted maps (e.g. Google Maps, Mapquest) and the terms of use for your map base layer must be compatible with our CC-BY 4.0 license.  Note: if you created the map in a software program like R or ArcGIS, please locate and indicate the source of the basemap shapefile onto which data has been plotted. If your map was obtained from a copyrighted source please amend the figure so that the base map used is from an openly available source. Alternatively, please provide explicit written permission from the copyright holder granting you the right to publish the material under our CC-BY 4.0 license. Please note that the following CC BY licenses are compatible with PLOS license: CC BY 4.0, CC BY 2.0 and CC BY 3.0, meanwhile such licenses as CC BY-ND 3.0 and others are not compatible due to additional restrictions.  If you are unsure whether you can use a map or not, please do reach out and we will be able to help you. The following websites are good examples of where you can source open access or public domain maps: * U.S. Geological Survey (USGS) - All maps are in the public domain. (http://www.usgs.gov) * PlaniGlobe - All maps are published under a Creative Commons license so please cite “PlaniGlobe, http://www.planiglobe.com, CC BY 2.0” in the image credit after the caption. (http://www.planiglobe.com/?lang=enl) * Natural Earth - All maps are public domain. (http://www.naturalearthdata.com/about/terms-of-use/)

Additional Editor Comments (if provided):

Reviewers' comments:

Reviewer's Responses to Questions

**Comments to the Author**

1. Does this manuscript meet PLOS Mental Health’s publication criteria?

Reviewer #1: Yes

Reviewer #2: Yes

2. Has the statistical analysis been performed appropriately and rigorously?

Reviewer #1: N/A

Reviewer #2: N/A

3. Have the authors made all data underlying the findings in their manuscript fully available (please refer to the Data Availability Statement at the start of the manuscript PDF file)?

Reviewer #1: Yes

Reviewer #2: No

4. Is the manuscript presented in an intelligible fashion and written in standard English?

Reviewer #1: Yes

Reviewer #2: Yes

Reviewer #1: This clearly written manuscript addresses the important topic of the perceptions of peer group leaders on their experiences delivering a mental health intervention in Tanzania. Given the recent increase of lay health worker-delivered mental health interventions in both LMIC and HIC settings, and particularly the focus on having these delivered by “peers” this is a timely and relevant focus. The authors conducted a qualitative sub-study comprised of in-depth interview data from the 25 Peer Group Leaders (PGLs) who delivered a mental health intervention for youth living with HIV in a trial conducted in Tanzania. There are some important findings about the motivators and challenges of working as a PGL, and some good implications for future programming. However, there are some areas that could benefit from further clarification and/or analysis. Although the study is stated to be based on the CFIR Implementation Science Framework, very few of the domains of this framework appear to have been addressed in the interviews and the Results are not presented in a way that shows how they relate to this framework. Further work is needed to specify and flesh-out sub-themes in the data, and to elucidated any sub-group differences. Specific comments and questions by section are given below.

Abstract:

1. The authors should mention the number of PGL interviews that were conducted for the study in the abstract.

Introduction:

2. Line 46: The statement that 62.5% of mental disorders occur before age 25 needs clarification. Does this mean that most people with mental illness begin to experience their mental disorders before the age of 25, or something else?

3. Lines 57-59: Is the first sentence presenting findings from a single local study or national data?

4. Line 63: What is the definition of “professional” in this sentence about the mental health workforce?

Methods:

5. Lines 98-99: The authors should specify whether SYV was developed originally for the study or is based/adapted on an evidence-based model implemented elsewhere. The multiple techniques being used (CBT, IPT, Motivational Interviewing) makes me wonder about the suitability for delivery by lay health workers with only a 2-week training.

6. Lines 98-99: The authors should also specify here the age range of the YLWH included in the SYV trial.

7. Lines 179-187: If this study was really informed by the CFIR framework, why were more of the CFIR domains not investigated in the interviews and analysis?

8. Lines 197-200: Was the Tanzanian qualitative researcher involved in the full coding and analysis, or just this double-coding of 8 transcripts? This is not clear.

9. Line 209: I am noting that COREQ includes criteria for reporting qualitative research not for conducting it.

Results:

10. Overall: It would be better to organize the Results according to your CFIR conceptual framework, instead of just the classic and simplified Motivators and Challenges themes. More sub-themes are also needed to more clearly flesh out the themes. Disparate sub-themes seem to be lumped together in the results. The whole analysis would be improved by fleshing out and specifying sub-themes.

11. Line 248: Applying skills learned to their own daily lives, for example, seems like a separate theme than “confidence”.

12. Lines 291-294: It is not clear from this quote how this effect stems from relationships with colleagues (the theme of this section).

13. Line 343: The theme “inclusivity and collaboration” is included in the Challenges section (I believe), but is this a challenge? Should this be labeled “lack of inclusivity and collaboration”?

14. Lines 371-37: This information on increased respect and recognition within the community seems that it should be moved up with the benefits and facilitators for the PGLs individually.

15. Lines 379-389: It is not clear why teaching competence and supervision are lumped together. These seem like separate sub-themes.

16. Line 408: This theme about career advancement seems that it would be better placed above as it is a concern of the PGLs about themselves and their future.

17. Were there any sub-group differences in the data, for example male vs. female PGLs? These should be analyzed and results presented.

Discussion:

18. Lines 459-467: It is not clear how the citations in this paragraph relate to the statements made here that seem to be about the results of the current study, not from the literature.

19. Lines 537-538: How will this recommendation affect cost and sustainability of this intervention in health systems?

Tables and figures:

20. The second column of the supplementary codebook file does not seem to add much as it is mainly just a repetition of the in-depth interview guide.

Reviewer #2: This is an important paper with useful lessons for providers of similar interventions. The role of PGL clearly has a substantial effect on many of those interviewed.

1. In reference 1 (The UNAIDS 2022 data report) I couldn’t find the specific information cited here: 150,000 YLWH in Tanzania, 100,000 young women living with HIV. The paper says “This disparity is evident in HIV prevalence rates, with adolescent young women having a rate of 1.7, nearly twice the rate of adolescent young men at 0.9”. I don’t know what this means. A prevalence is not a rate, and 1.7 is neither a rate or a prevalence, it’s just a number. What is it you want to say? What occurs at a rate of 1.7 what, per what?

2. I’m not clear about the relevance of the information in the introduction that youth who found out their HIV status independently had poorer ART adherence. All the youth in this study already know their HIV status. However, if this point is relevant, you might be interested in the paper below which found that youth in Zimbabwe with symptoms of anxiety/depression who had not told anyone their HIV status had poorer viral suppression (and also that many youth knew their status although their parents did not think they they knew it).

Simms et al (2021) Risk factors for HIV virological non-suppression among adolescents with common mental disorder symptoms in Zimbabwe: a cross-sectional study. JIAS https://doi.org/10.1002/jia2.25773

3. It is good to learn that PGLs have benefitted in confidence, friendships, mental health and self-acceptance from their work. However, the discussion should address whether this means that perhaps at the start of the program, some of them should not have been supervising groups of adolescents. Despite the name, the PGLs are not peers of the youth. They are adults who are taking responsibility for youth in the groups. It is definitely important to provide support systems for the PGLs to manage emotional discomfort caused by the role, as you said in the discussion.

4. The section about social dynamics between PGLs is unclear. The quote on lines 326-328 doesn’t seem to be about issues being escalated, but the text says that it is.

Minor comments

1. What is Duke Box?

2. I recommend writing dates in full, eg. 25 October 2021 instead of 25/10/2021.

3. The abstract says the PGLs were aged 23-29 but the text says they were aged 23-31.

**Do you want your identity to be public for this peer review?** For information about this choice, including consent withdrawal, please see our Privacy Policy

Reviewer #1: No

Reviewer #2: No

---

## [Decision Letter · Decision Letter 1]

8 Oct 2025

PMEN-D-24-00520R1

Peer-led interventions: exploring the peer group leader experience of delivering Sauti ya Vijana, a group-based mental health intervention for youth living with HIV in Tanzania

PLOS Mental Health

Dear Dr. Agina,

Thank you for submitting your manuscript to PLOS Mental Health. After careful consideration, we feel that it has merit but does not fully meet PLOS Mental Health’s publication criteria as it currently stands. Therefore, we invite you to submit a revised version of the manuscript that addresses the points raised during the review process.

We look forward to receiving your revised manuscript.

Kind regards,

Kizito Omona, PhD

Academic Editor

PLOS Mental Health

Journal Requirements:

Additional Editor Comments (if provided):

Address the comments raised by the reviewers

Reviewers' comments:

Reviewer's Responses to Questions

**Comments to the Author**

Reviewer #1: (No Response)

Reviewer #2: All comments have been addressed

publication criteria?

Reviewer #1: Yes

Reviewer #2: Yes

3. Has the statistical analysis been performed appropriately and rigorously?

Reviewer #1: N/A

Reviewer #2: Yes

4. Have the authors made all data underlying the findings in their manuscript fully available (please refer to the Data Availability Statement at the start of the manuscript PDF file)?

Reviewer #1: Yes

Reviewer #2: No

5. Is the manuscript presented in an intelligible fashion and written in standard English?

Reviewer #1: Yes

Reviewer #2: Yes

Reviewer #1: The authors have generally done a very good job responding to the editor and reviewer comments and making relevant changes in the manuscript. The additional clarity on how CFIR was used in the qualitative data collection, analysis, and reporting of results is appreciated. There are a few remaining minor concerns:

• The response letter (response to Reviewer #1, response 1) indicates an N of 23 interviews, whereas elsewhere an N of 25 is indicated.

• The abstract should be edited to better reflect the new thematic organization of results around the CFIR Individuals Involved and Inner Setting themes. The abstract still reflects the previous organization around benefits and challenges.

• I would suggest that the information provided in the response letter on how the data were interrogated for sub-group differences and the lack of sub-group differences found in the data (response to Reviewer #1, response 16) should be included in the Methods section of the manuscript (Data analysis).

• One could argue that the sub-theme around community perceptions of the PGLs would fall under the Outer Setting domain of CFIR. The authors may want to consider separating this out in that way, along with any other Outer Setting findings.

Reviewer #2: Thank you, all my comments have been addressed.

One very minor point - the prevalence of HIV in girls and young women (1.3%) is more than twice the prevalence in boys and young men (0.6%), not almost twice.

**Do you want your identity to be public for this peer review?** For information about this choice, including consent withdrawal, please see our Privacy Policy

Reviewer #1: No

Reviewer #2: No

---

## [Editor Report · Decision Letter 2]

24 Nov 2025

Peer-led interventions: exploring the peer group leader experience of delivering Sauti ya Vijana, a group-based mental health intervention for youth living with HIV in Tanzania

PMEN-D-24-00520R2

Dear Dr. Agina,

We are pleased to inform you that your manuscript 'Peer-led interventions: exploring the peer group leader experience of delivering Sauti ya Vijana, a group-based mental health intervention for youth living with HIV in Tanzania' has been provisionally accepted for publication in PLOS Mental Health.

Best regards,

Kizito Omona, PhD

Academic Editor

PLOS Mental Health

Thank you for addressing the comments raised by the reviewers